# Chitosan Nanoparticles in Atherosclerosis—Development to Preclinical Testing

**DOI:** 10.3390/pharmaceutics14050935

**Published:** 2022-04-25

**Authors:** Pornsak Sriamornsak, Crispin R. Dass

**Affiliations:** 1Department of Pharmaceutical Technology, Faculty of Pharmacy, Silpakorn University, Nakhon Pathom 73000, Thailand; sriamornsak_p@su.ac.th; 2Pharmaceutical Biopolymer Group (PBiG), Silpakorn University, Nakhon Pathom 73000, Thailand; 3Academy of Science, The Royal Society of Thailand, Bangkok 10300, Thailand; 4Curtin Medical School, Curtin University, Bentley 6102, Australia; 5Curtin Health Innovation Research Institute, Bentley 6102, Australia

**Keywords:** chitosan, nanoparticle, atherosclerosis, cardiovascular, drug

## Abstract

Chitosan is a natural biopolymer that is present in an abundant supply in sources such as crustacean shells, mushrooms, and insect exoskeletons. It can be used to make a variety of types of drug formulations and is generally safe to use in vivo; plus, it has inherent cholesterol-reducing properties. While an abundance of papers has tested this biopolymer in nanoparticles in cancer and diabetes research, there is a lag of usage, and hence the paucity of information, in the area of cardiovascular research, specifically in atherosclerosis, the topic of this review. This review highlights some of the deficiencies in this niche area of research, examines the range of chitosan nanoparticles that have been researched to date, and proposes several ways forward to advance this field. Nanoparticles used for both diagnostic and therapeutic purposes are reviewed, with a discussion on how these nanoparticles could be better researched in future and what lays ahead as the field potentially moves towards clinical trials in future.

## 1. Atherosclerosis—A Brief Introduction

Atherosclerosis is a disease most typically characterised by thickening and/or hardening of the arteries caused by an accumulation of plaque in the lumen of an artery [1,2]. As such, the blood supply to the heart muscle, brain, or peripheral tissues may be compromised, which may be fatal if left unchecked [3,4,5]. Atherosclerosis is responsible for a majority of cardiovascular disease deaths [6,7]. Risk factors or triggers for the disease include high cholesterol, triglyceride and saturated fat levels (mainly through the diet), smoking, high blood pressure, obesity, and reduced or nil physical activity [8,9,10]. 

Of these, we hear most of the harmful effects of cholesterol. It was Virchow, more than 100 years ago, that discovered that atheroma contained a yellow fat-containing matter, later to be identified as cholesterol by Windaus [11]. The current understanding of the development of atherosclerosis is the oxidation of lipids and lipoproteins that enhance the development of plaques [12]. The reader is directed to other excellent recent review articles that comprehensively describe the complex process of atherosclerosis development [13,14,15]. Apart from cholesterol, plaques are composed of fatty molecules such as phospholipids, calcium, and other constituents, remain stable throughout an individual’s life, or they become unstable and can grow to such a size that they pose a health risk from the stenosis (partial blockage of the artery), with a growing chance of disruption. Calcification of plaques may render the vessel rather incapable of regulating blood flow efficiently due to reduced flexibility within the vessel walls when compared to healthy patent vessels.

An atherosclerotic plaque can rupture suddenly and without much warning, leading to fatal events due to severe ischaemia [16,17]. While the good news is that the disease can be largely prevented through proper diet and activity in the early years of one’s life [18,19,20,21], once it sets in, it is quite hard to eradicate and often easy to exacerbate, even to the point of becoming severely life-threatening. Thus, newer methods for imaging (diagnosing) and managing (treating) this disease are being sought [22,23]. 

The disease process begins when low density lipoprotein (LDL; usually termed ‘bad cholesterol’) in the bloodstream passes through the gaps between endothelial cells and enters the intima of the arterial wall. The LDL particles, which are not harmful if they are left as is, unfortunately, are modified via oxidation into oxidized LDL (oxLDL). The immune response evoked by the oxidation of LDL cholesterol urges the endothelial cells near the inflammatory site to recruit monocytes from the bloodstream, which enter the intima (Figure 1). Once within the intima, the monocytes differentiate (or become specialised) into macrophages in the arterial wall and ingest the oxLDLs through surface-based scavenger receptors [24]. In a further step towards atherosclerosis, these macrophages convert to foam cells, lipid-laden phagocytic cells that contribute to the fatty streak architecture seen in diseased arterial walls. This sequesters cholesterol within the artery wall. The maturation of fatty streaks into more advanced plaques produces lesions that are usually covered with a fibrous cap composed of migrated smooth muscle cells (SMCs) and extracellular matrix (ECM) proteins such as collagen [25]. 

The migration of SMCs responds to a chemical signal produced during the accumulation of oxLDL, foam cells, and debris. This results in the formation of a fibrous cap, a layer of connective tissue that covers an atherosclerotic plaque and shields the lesion from the vascular lumen. The fibrous cap encloses a lipid-rich necrotic core composed of oxLDLs, cholesterol and apoptotic or necrotic cells that are unable to obtain sufficient nutrients for survival. As the inflammatory process progresses, apoptosis (cell death that occurs normally) and matrix degradation by matrix metalloproteinases (MMPs) become apparent. As inflammation escalates, accompanied by persistent foam cell recruitment and the ever more necrotic environment within the atherosclerotic plaque, further development and maturation of the atherosclerotic lesion ensue. This assists in the enlargement of the lipid-rich core and the thickening of the fibrous cap.

## 2. The Development of an Atherosclerotic Plaque—A Closer Look at the Cells Involved

One of the very early stages of atherosclerosis is the inflammation of the endothelial lining, which transits through various stages. This involves impairment of vasoprotective nitric oxide (NO) production and increased expression of cell adhesion molecules (CAMs), which facilitate monocyte recruitment and their eventual differentiation to macrophages [26]. Increased expression of vascular cell adhesion molecule-1 (VCAM-1) and P-selectin plays an important role in endothelial inflammation, as it is widely known now that these molecules are part of the recruitment ‘drive’ attracting cells such as monocytes and other leukocytes. VCAM-1 expression at the surface of vascular endothelial cells in areas that are likely to develop atherosclerosis (such as aortic arches) precedes the accumulation of macrophages in the arterial intima, as the latter relies on the former [27,28]. 

When the metabolism of low-density lipoprotein (LDL) is abnormal, as in the onset of atherosclerosis, the increased LDL in the blood will penetrate directly into the inner subcutaneous layers of the vascular wall through the loose gaps between the endothelial cells [29]. The endothelial LDL is then oxidised to produce oxLDL [30]. This oxLDL, typified by oxysterols such as 7β-hydroxycholesterol and 27-hydroxycholesterol [31], is phagocytosed by macrophages, which thereby accumulate large amounts of cholesterol and become foam cells. If left unchecked, foam cells continue to accumulate, forming fatty striations in affected arterial walls that eventually lead to atherosclerosis.

Atherosclerosis is essentially a chronic inflammatory disease [32]. As part of such inflammation, circulating monocytes migrate into the endothelium and differentiate into macrophages when exposed to a rich source of growth factors and pro-inflammatory cytokines [33]. Early atherosclerotic lesions are typified by an increased number of macrophages and the formation of macrophages that have engorged lipid droplets (these cells are called foam cells) [34]. Exogenous foam cells have the same cellular origin, same composition, and a cytological morphology that is very similar to that found in microscopic atherosclerotic plaques [35], making them an ideal In vitro (cell culture) model in which to study specific elements of atherosclerosis disease progression (example intracellular lipid intake and metabolism) and potential therapy (example cholesterol efflux). More mature lesions have pools of extracellular lipid and foam cells that have collected at the site (Figure 1) from macrophages and VSMCs in the artery wall [36]. These lesions are contained by a fibrous capsule made up of VSMCs that have migrated to underline the above VECs [37]. Stable plaques are characterised by a relatively small lipid core enclosed by a thick fibrous cap, whereas unstable plaques are mostly characterised by a large lipid core covered by a thin fibrous cap and are prone to rupturing at any time [36]. 

Macrophages, precursors of which are circulating monocytes, are the most abundant cellular components in vulnerable plaques. When the metabolism of LDL is abnormal, the increased LDL in the blood will extravasate into the inner subcutaneous [layer of the vascular wall through the loose gaps between activated endothelial cells. Then, the endothelial LDL oxidizes to produce mildly modified oxidized low-density lipoprotein (oxLDL). oxLDL is phagocytosed by macrophages, causing macrophages to accumulate large amounts of cholesterol and become foam cells, which exist amongst migrating VSMCs. Foam cells continue to accumulate, forming fatty striations in the arterial wall that eventually lead to the development of atherosclerosis typified by rupture-prone plaques that are normally covered with a fibrous capsule.

## 3. Chitosan—A Promising Biopolymer That Has Potential in Atherosclerotic Disease Management

Chitosan is a linear polysaccharide composed of randomly distributed β-(1–4)-linked 2-amino-2-deoxy-d-glucopyranose (deacetylated units) and 2-acetamido-2-deoxy-d-glucopyranose (acetylated units) [38]. Chitosan is derived from chitin, a highly abundant natural biopolymer with a high cationic potential [39]. Chitin is extracted from the exoskeleton of many living organisms, including crabs, shrimps, insects, and fungi. Chitosan possesses various highly desirable features, which include biocompatibility and versatility as far as formulation techniques are concerned. These characteristics make chitosan a highly sought after biomaterial with technological applications in cosmetics [40,41], pharmaceutical [42,43], and biomedical [44,45] industries. However, despite all this, as this review will highlight, there is a deficiency of usage of chitosan in the formulation of nanoparticulate technologies that can be used for, and are being used for, atherosclerosis research (diagnostics, therapeutics, and theranostics). Of significant importance is the finding that chitosan per se possesses the ability to reduce cholesterol levels [46,47], which further adds to its arsenal as an attractive polymer in the development of drug delivery systems for the treatment of atherosclerosis.

As chitosan has various beneficial effects on human health, for instance, in bone healing [48,49], it makes it quite an ideal biomaterial to employ in the drug delivery research sector. Various labs, including ours [50,51,52,53,54,55,56,57,58,59,60,61,62,63], have shown that chitosan is capable of encapsulating various types of therapeutic agents such as small molecule drugs to exploratory biologicals such as proteins and even DNA in delivery platforms of various sizes. As chitosan can even be ingested with no complications, this fact further cements the usefulness of this biopolymer in drug delivery R&D. Chitosan NPs have been formulated in the past using various methods, not limited to complex coacervation [64] and ionotropic gelation [65], both simple enough to carry out in most labs.

## 4. Chitosan Nanoparticles Used in Atherosclerosis Research

Chitosan can be used as-is or as a chemically-modified form, such as a commonly used form in hydrophobically modified glycol chitosan (HGC). While not reviewed here, chitosan can also be derived into charged varieties, such as *N*-trimethyl chitosan chloride [66], which has been used, for instance, in diabetes [67] and cancer [68] research in drug delivery attempts. HGC NPs are biodegradable, have low immunogenicity and are adaptable to carry a variety of therapeutic agents in vivo [69,70]. Depending on the formulation technique used, this biopolymer can be relatively inexpensive and easy to use, as we [60,61,62,71,72,73] and others [74,75] have established previously. We now review the collection of articles in the literature that have used chitosan-based NPs in atherosclerosis research (none has yet been tested clinically, so the discussions revolve around cell culture and preclinical animal models testing)—that is, tested as a diagnostic, therapeutic or both diagnostic and therapeutic (theranostic) platform. A summary of these studies is provided in Table 1 (in vitro evaluation) and Table 2 (in vivo evaluation).

One of the earliest studies, published in 2008 [76], was by a team who chemically conjugated an atherosclerotic plaque-homing peptide (‘AP peptide’, CRKRLDRNC) to HGC NPs. This peptide was derived from a process called biopanning—during which phage populations (from phage display libraries) are exposed to targets such as those found on VECs, macrophages and VSMCs that are found in atherosclerotic plaques, to selectively capture phages that bind to specific targets. As described in Park et al. [76], after several rounds of binding, washing, elution and amplification, the originally diverse phage population is reduced down to a phage population with a propensity to bind to the desired target. The authors used this technique as it had been before to identify differences in the molecular moieties expressed on the surfaces of VECs of differing tissues and even tumours [91,92]. The advantage of phage display technology is the easy discovery of peptides binding selectively to molecular targets, thus reducing the time and labour invested to generate lead candidates that have potential as imaging agents.

Thus, tissue targeting, such as that to atherosclerotic plaques, may be possible by conjugating NPs to antibodies or peptides known to specifically bind to targets in specific disease tissues. Using bovine aortic endothelial cells (BAECs) as a model, Park et al. [76] were able to show both avid binding and more intracellular delivery of the NPs in these cells. The authors proposed that the activated BAECs may be expressing interleukin-4R (IL-4R) (which is induced by TNF-α exposure) on the surfaces of cells, suggesting that the AP peptide conjugated to HGC conjugate retained a binding affinity for IL-4R on activated BAECs. Thus, the NPs bound more selectively to TNF-α-activated BAECs than to unstimulated cells under both static and dynamic flow conditions. In vivo, the AP-tagged HGCCy5.5 nanoparticles bound better to atherosclerotic lesions in an LDL receptor-deficient (LDLr-/-) atherosclerotic mouse. NPs bound to the luminal surfaces of atherosclerotic lesions and inside the lesions, which are composed mainly of VSMCs and macrophages. While the authors expected that their AP-tagged HGC nanoparticles containing drugs would soon be developed for therapeutic purposes, no such further development has occurred to date, 13 years later. This seems to be the trend for other similar technologies discussed below, despite the relatively promising data obtained preclinically, some more than five years ago.

Complex coacervation-prepared NPs composed of just chitosan and plasmid (expressing cholesteryl ester transfer protein, CETP) DNA were tested more than a decade ago [88]. As our lab has demonstrated [50,55,56], this is one of the easiest methods to formulate NPs and depends essentially on the electrostatic attraction between the cationic chitosan and an anionic counterpart such as dextran sulphate [93,94] or oligonucleotides [95,96]. This is akin to the use of cationic liposomes to bind DNA [97], and one of the issues could be an inefficient release of the anionic DNA from the polycationic chitosan chains, a concern with other biopolymeric systems such as gelatin-based delivery systems [98]. In the Yuan et al. [88] study, the area of aortic lesions and intimal thickening were greatly reduced. One of the major advantages of an NP system such as this is the relative ease of manufacture, sometimes using simple instrumentation such as a speed-adjustable vortex mixer, with the added advantage of the use of cost-effective components and equipment that most labs would have inhouse.

However, unlike other uses for gene delivery for other disease indications such as cancer, this technology has not really been further investigated or developed in atherosclerotic disease. It is important to point out, though, that the field of gene therapy has undergone several changes since the early clinical studies and showed side effects that even led to the death of some patients [99]. This is a far cry from the original promise the technology held [100,101], which has since lost some of its original allure. The intricacies of targeting a gene therapy construct such as plasmid DNA or viral vectors (such as adeno-associated vectors, AAVs) through the bloodstream to the target site, which is essentially part of that same bloodstream (that is a junction of the artery that is plagued with a lesion), is manifold. Not only does the delivery system have to overcome the mechanical difficulties of the blood flow, but also the intrinsic nature of its cargo (that is, DNA), which is naturally prone to degradation by nucleases In vivo. These, plus the fact that the cargo is sometimes 100 kDa in size, make this technology a difficult one to bring to fruition.

Stabilin-2 is a glycoprotein highly expressed on macrophages, and VSMCs and VECs make atherosclerotic plaques [102]. Stabilin-2 acts as a receptor for advanced glycation end-products (AGEs) [103], chemical species that accumulate within atherosclerotic plaques of vessels. VECs and other cells constituting initial atherosclerotic events such as macrophages and foam cells of fatty streak lesions accumulate AGEs [104], which precede the development of maturing atherosclerotic plaques and then advanced lesions. In their study, Lee et al. [89] identified a peptide (CRTLTVRKC), termed S2P, from a phage display assay that was capable of binding avidly to stabilin-2. It took four rounds of selection to derive the peptide. S2P conjugated to HGC NPs was efficiently delivered to atherosclerotic plaques, becoming enriched in plaque regions. Interestingly, the work by this team was built on their findings reported in the same paper [89], which found that stabilin-2 was strongly expressed in VSMCs and VECs, as well as macrophages, of atherosclerotic lesions, suggesting that the protein could usefully serve as a target in atherosclerosis. Furthermore, the group established that injected S2P peptide homed to lymph nodes and spleen, that is, organs expressing stabilin-2, and colocalised with mouse stabilin-2 in the VECs of sinusoids within these tissues. Of slight concern was the finding that S2P localised strongly in the liver, which the authors attributed to the liver’s uptake of hydrophobic and low molecular-weight compounds.

Epigallocatechin gallate (EGCG) is a polyphenolic compound found in green tea, which has vasculoprotective effects [105]. In humans, EGCG inhibits endothelial dysfunction and improves brachial artery dilation in patients with atherosclerosis [106]. However, it has poor stability and bioavailability in humans [107]. An attempt to nanoencapsulate EGCG in chitosan/polyaspartic acid NPs led to favourable stability In vitro (simulated gastric and intestinal fluids) and reduction of plaque load in vivo [77]. In this study, the average ratios of lipid deposit area for EGCG NP-fed rabbits and EGCG-fed rabbits were 16.9 and 42.1, respectively, which showed that the EGCG NPs were significantly more effective against atherosclerosis compared with free EGCG. This was partially due to better protection of the phenolic compound within the NPs, given its low stability in water and physiological fluid because it readily undergoes oxidation, degradation, and polymerisation [108]. EGCG is unstable in sodium phosphate buffer (pH 7.4), where 80% of it is lost merely within 3 h [109,110]. In fact, the efficacy of EGCG NPs against rabbit atherosclerosis was close to that of simvastatin, a clinically used drug, which was in itself a remarkable achievement. Corresponding to these beneficial effects In vivo, the EGCG NP and EGCG dampened the levels of TG, TC, HDL-C, and LDL-C by 52, 55, 27, and 65% and 19, 26, 23, and 33%, respectively. Thus, the NPs showed efficacy in controlling levels of lipids that are detrimental in instigating and maintaining atherosclerosis. Bodyweight changes were not noted in the NP group, suggesting no gross toxic effects were evoked in treated mice.

Superparamagnetic iron oxide NPs (SPIONs) formulated from a cationically derivatised chitosan were surface-modified with antibodies against either VCAM-1 or p-selectin [78]. In vitro studies confirmed the specific interaction of anti-VCAM-1 antibodies bound to the surface of SPIONs with aortic endothelial cells (AECs) derived from db/db mice, that is, cells represented those found in a state of inflammation typical of atherosclerotic plaques. Furthermore, these cells are usually inflamed as a complication of diabetes and, as mentioned above, constitute the early stages of plaque initiation before monocytes are attracted through VCAM-1 expression. In an ex vivo analysis, harvested aortic arch specimens with part of the descending aorta and brachiocephalic artery aortic ring present were obtained from ApoE/LDLR-/-mice with endothelial dysfunction and incubated with SPION-CCh-anti-VCAM-1 nanoparticles. The presence of SPIONs was confirmed by magnetic resonance imaging (MRI), and strong binding to the surface of the aorta was demonstrated. One finding of slight concern, and one that would need to be evaluated in an In vivo model in future, was that agglomerates of NPs were localised in the outer part of the aorta too, which may mean that the antibodies used are not sufficiently selective towards inflamed endothelium and may also interact with perivascular tissue. This could be well due to the non-selective binding of the SPIONs to the outer surface, perhaps due to the way the tissue was positioned during the assay period.

Using a pegylated version of chitosan, Hirpara et al. [79] encapsulated a long-circulating form of NP ferrying rosuvastatin. Statins are 3-hydroxy-3-methylglutaryl coenzyme A (HMG CoA) reductase inhibitors proven to reduce total and LDL cholesterol [111]. In addition to the beneficial cholesterol-reducing effects, statins improve endothelial function, aid in the stabilisation of atherosclerotic plaques, and inhibit inflammatory and thrombogenic responses in arterial walls that enhance the risk of developing full-blown atherosclerosis [112]. To date, marketing surveillance has shown that long term statin therapy is generally well-tolerated, and this drug class is one of the frontline drugs used for atherosclerosis management clinically. In the study performed by Hirpara et al. [79], In vitro drug release demonstrated a sustained rate of release in phosphate buffer. In line with the pharmacokinetic (PK) study, in a hyperlipidaemic rat model, NPs enhanced the lipid-lowering capability of rosuvastatin compared to free drugs. The authors proposed that it was the PEGylation of NPs which helped to avoid opsonisation from the blood, allowing the NPs to circulate longer in the circulation, maintaining sustained release.

Ferric-oxide-containing NPs made from chitosan, perfluorohexane (PFH), poly(lactic-co-glycolic acid) (PLGA), and dextran sulphate (DS) were tested in combination with low-intensity focused ultrasound (LIFU) irradiation to locate atherosclerotic lesions in mice [80]. These NPs carried ferric oxide, which could be picked up readily with MRI In vivo. In vitro, NPs selectively bound to activated macrophages via targeting class A scavenger receptors (SR-A) with the DS moiety. This was mirrored in an ex vivo aortic arch plaque model where NPs selectively bound to the lesion sites. In vivo, not only did the NPs target plaques, but they led to apoptosis of activated macrophages with the use of LIFU technology. The study built upon the expertise reported in an earlier paper by Zhu et al. [113], who successfully developed a novel NP that combined LIFU with an “explosion effect” within cells that was caused by an acoustic droplet vaporisation (ADV) phenomenon. In the process of ADV, a series of violent dynamic processes such as oscillation, expansion, contraction, and even the collapse of tiny bubbles in the liquid under the action of sound waves are generated, thereby inciting chemical reactions luminescence and subharmonics. These processes essentially damage the ultrastructure within cells, culminating in apoptosis and thereby death and removal of diseased cells [114].

For the past decade, nanotechnology-based RNAi therapeutic platforms have shown great promise for various disease applications, though not many such systems have been tested in the field of atherosclerosis research. A pegylated form of chitosan [81] was used to formulate NPs carrying mir-33, which is capable of reducing ATP binding cassette subfamily A member 1 (ABCA1). Ionic gelation mediated via tripolyphosphate (TPP) crosslinking, a relatively straightforward technique, was utilised for NP manufacture. In this technique, an ionic interaction between the cationic charge of primary amine groups of the chitosan with the anionic charge present in the nucleotides making up the miRNA mimics occurs, where TPP is used to stabilise the miRNA mimic–chitosan and chitosan–chitosan interactions. In vivo, a decreased reverse cholesterol transport (RCT) to the plasma, liver, and faeces was noted. Furthermore, when cholesterol efflux-promoting miRNAs were delivered with these NPs, ABCA1 expression and efflux into the RCT pathway were improved. In vitro, the efficiency of intracellular delivery of different formulations of these NPs to mouse macrophages was performed through an evaluation of the extent of cholesterol efflux to apolipoprotein A1 (ApoA1). NPs were also found to have no toxicity towards these cells in culture and were long-circulating owing to the pegylated moieties on their surface. Specifically, compared to empty NPs, mice injected with miR223-chNPs- or miR206-chNPs-loaded macrophages demonstrated increased plasma cholesterol (27% and 31% for miR223-chNPs, and 35% and 46% increase for miR206-chNPs over 24 and 48 h, respectively). Effluxed cholesterol uptake in the liver increased by 27% from miR223-chNPs and by 40% from miR206-chNPs-loaded macrophages. Treatment with miR206-chNPs also increased total faecal cholesterol excretion by 45% and 60% at 24 and 48 h, respectively, confirming that the NPs indeed were efficacious.

CD47-targeted NPs were formulated in an attempt to deliver selectively to vascular endothelial cells (VECs) [82]. As the authors note, in atherosclerotic patients, high CD47 expression is commonly found in plaques. The nanoadjuvant approach employed by Yu et al. [82] relies on the fact that NP-encapsulated or -adsorbed antigens are the first targets of dendritic cell and macrophage phagocytosis and constitute an important step to achieving an effective immune response [115]. Nano-encapsulated antigens are abundant in antigen-presenting cells (macrophages, dendritic cells) and are strongly targeted to them. Furthermore, they are highly immunogenic. Fluorescently-labelled NPs were able to adsorb to VECs In vitro, though no attempt was made to examine whether NPs were, in fact, inside cells. In atherosclerosis-prone apolipoprotein E-deficient (ApoE-/-) mice, these NPs localised around the aortas, demonstrating the selective mode of delivery. Summarily, this study established that atherosclerotic plaque clearance was promoted by blocking the expression of CD47 on the surface of atherosclerotic tissues by antibodies that were adsorbed on chitosan NPs. Additionally, these NPs, when combined with foam-like cells, exerted a negative immunomodulatory effect on exogenous immune cells, thereby further enhancing the inhibitory effect that was observed on atherosclerotic plaques in this study. In vivo, the chitosan-containing nanoadjuvants were shown to mediate the activation of type 1 regulatory (Tr1 or, more specifically, CD44þ Foxp3-Tr1) cells by exogenous foam cells. Tr1 cells inhibit NLRP3 ((NOD-, LRR- and pyrin domain-containing protein 3) activation in macrophages [116]. NLRP3-related inflammatory factors are highly expressed during the formation of microscopic plaques and are not only involved in early arterial plaque formation but are also in inflammatory responses and immune regulation In vivo [117,118]. Thus, a major inducer of atherosclerosis progression was regulated negatively. As a side issue, for most types of chitosan-containing NPs, these hyaluronic-acid (HA)-containing ones were both easy to manufacture and were non-toxic to the macrophage cells.

As mentioned above, dysfunction at the endothelial layer leads to a build-up of the oxidized form of LDL in the intimal layer of the artery, whereby local inflammation culminates in the excess generation of reactive oxygen species, ROS [119]. ROS are ubiquitous in animals, given that they are byproducts of aerobic metabolism. As we know, oxygen’s outer valence shell consists of six electrons, resulting in two unpaired electrons. Thus, various ROS can be generated by increasing the electrons around the oxygen. Common examples include hydrogen peroxide (H_2_O_2_), superoxide anion (•O_2_^−^), peroxide (•O_2_^−2^), and hydroxyl radical (•OH) [120,121].

Macrophage membrane coated ROS-responsive NPs (MM-NPs) were formulated with oxidation-sensitive chitosan [83]. Using these coated NPs, this group was able to demonstrate results suggesting that these NPs may provide an effective strategy to escape macrophage clearance, reacting to ROS for enhancement of atorvastatin release. MM coating significantly increased NP circulation time by decreasing uptake by the reticuloendothelial system (RES). Most importantly, atorvastatin-carrying MM-coated NPs decreased plaque size at the aortic arch of ApoE-/-mice. They also stabilised plaques due to the increased presence of VSMCs and increased collagen.

Chen et al. [84] formulated chitosan NPs carrying rosuvastatin and tested them in hypercholesterolemic rabbits. Major findings from this study include the fact that rosuvastatin-containing NPs were found to be effective in maintaining the blood calcium level to near resting levels, blood interleukin-1 (IL-6) levels to resting levels, and reducing both total and LDL cholesterol levels. Rosuvastatin-loaded chitosan NPs were found to be significantly effective in lowering blood lipid levels in comparison to pure statin. Apart from lowering blood calcium, it reduced the calcification of various valve tissues in test rabbits. When challenged in a PBS release study, NPs released 12% of the loaded drug in a burst mode over 5 h, and the remnant 88% over both gradual and burst modes (latter presumably due to NPs breaking) over 48 h.

In an interesting study where both rosuvastatin and lovastatin were formulated into chitosan NPs, researchers [85] were able to demonstrate that administration of these NPs led to reduced blood cholesterol (total and LDL) and serum uric acid. Uric acid has been associated with the development of atherosclerosis, and atherosclerotic plaque specimens display a higher concentration of uric acid than non-atherosclerotic control specimens [122], suggesting that serum uric acid might play a role in the aetiology of the disease. NPs containing only rosuvastatin or lovastatin were not as efficient as the dual drug-containing NP formulation In vivo, pointing to the combination being the better pharmacodynamic approach in this model. Large numbers of foam cells within plaques following cell infiltration and nuclear condensation were observed in the control group, while in the test group receiving NPs containing both rosuvastatin and lovastatin, no nuclear condensation or calcium deposits, nor cholesterol or fat deposits, were observed.

A common theme in several NPs discussed above is the reduction in inflammation-related events in atherosclerosis. Inflammatory events usually involve the generation of oxidation species such as ROS, which, as mentioned above, in atherosclerosis are produced by activated macrophages and are enriched in plaques [123]. Wu and colleagues [86] developed chitosan NPs containing both nanoceria and SPIONs for ROS-related theranostics targeting atherosclerotic lesions. While the group did not report on In vivo testing, their formulation was able to reduce ROS levels in activated macrophages, had a good contrast under MRI, and had little to no toxicity in cell culture.

Of the 25 known proteins that contain selenium, the so-called selenoproteins, 12 have redox activity and include enzymes such as glutathione peroxidase (GPx) and thioredoxin reductase (TrxR) [124]. Chitosan was used to coat the surface of selenium-containing NPs [90]. There were reduced atherosclerotic lesions in ApoE-/-mice after oral administration of NPs for 12 weeks. In ApoE-/-mice treated with different doses of selenium NPs for 12 weeks, total serum cholesterol, triglycerides and LDL-cholesterol levels were significantly lower, and HDL-cholesterol levels were significantly higher than those of the control cohort. These NPs reduced the lesion area by 31%. Intimal wall thickening was reduced by the NPs.

Most recently, a single-chain fragment variable (scFv) reactive to LDL (-) as a ligand was used to formulate surface-functionalised nanocapsules [87]. Its ability to attenuate LDL (-) uptake by primary macrophages and the progression of atherosclerotic lesions in LDLr-/-mice was studied. These NPs were internalised by human and murine macrophages. Tested macrophages exhibited lower LDL (-) uptake and reduced expression of interleukin-1B (IL1B) and monocyte chemoattractant protein-1 (MCP1) induced by LDL (-) when treated with this NP. The NP inhibited atherosclerosis progression without affecting vascular permeability or inducing leukocyte-endothelium interaction. Cells internalised NPs via phagocytosis, mainly through pinocytosis. In vivo, these NPs did not induce haemolysis or change leukocyte-endothelium interactions and vascular permeability. They also resulted in smaller atherosclerotic lesions with reduced lipid content at the aortic arch.

## 5. Future Perspectives

Atherosclerosis is the largest killer in the developed world and is fast becoming one in developing nations mainly due to the conversion of diets and lifestyles emulating that of the Western world. A total cure via way of medical intervention is still at arm’s length, and fervent research efforts are currently underway to find better and safer ways to manage the disease in its various manifestations and degrees of severity. The initial step for treatment mostly includes a reduction in risk through some changes in lifestyle, which include dietary and physical activity changes, and avoiding known risk factors such as smoking, in both its active and passive forms. However, when these changes fail short, either due to the non-compliance by patients or to the severity of the disease, clinical intervention is needed to avoid almost certain mortality. To this end, statins (HMG-CoA reductase inhibitors) are the choice of drug for the treatment of disease. However, the delivery of statins remains a big issue, as is finding better ways of imaging the disease. While it is clearly evident that there is scope for the use of chitosan for the formulation of NPs that could be of diagnostic and/or therapeutic use in atherosclerosis, as discussed in this review paper, not much has been performed with the technologies that were initially reported more than a decade ago. However, it is hoped that the more recent technologies, with better capabilities such as targeting moieties such as antibodies or ligands, may fare better in the fight against atherosclerosis with chitosan-based NPs. NPs encounter extremely complex physiological environments once administered In vivo, and in healthy individuals, a very potent and adept defence (immune) system that is actively recognising and clearing material foreign to the body. It is not a big surprise that despite the hype surrounding nanotechnology to date, very few nanoformulations coated with targeting ligands have passed phase III clinical trials. Most NPs, including those made from chitosan, are taken and removed by the RES (liver, lung, spleen, bone marrow, lymph nodes) before reaching the target tissues [125]. In the preclinical models reviewed above, there is no doubt that both imaging and reduction in the size of atherosclerotic lesions are feasible with these chitosan-based NP platforms. This area of research is still in its infancy, perhaps due to the fact that imaging of atherosclerotic lesions is more of a recent phenomenon. Certainly, the packaging of statin drugs such as rosuvastatin into chitosan NPs holds promise and is one avenue that warrants further investigation as the technology slowly meanders its way towards clinical trials. This progress is certainly made easier by the fact that toxicity is not a drawback for these chitosan-based biocompatible and biodegradable NPs, plus the numerous benefits of using chitosan. Better design of studies, such as the inclusion of histological evaluation of affected plaques and monitoring side effects in organs such as those of the RES (liver, for example) to complement serum analyses of cholesterol levels, are warranted moving forward. The importance of being able to produce scaled-up versions of such promising formulations in sterile environments and the need for these to be stable and transportable over large distances cannot be underemphasised as this will become important when NP formulations are transported to developing nations in future.

## Figures and Tables

**Figure 1 pharmaceutics-14-00935-f001:**
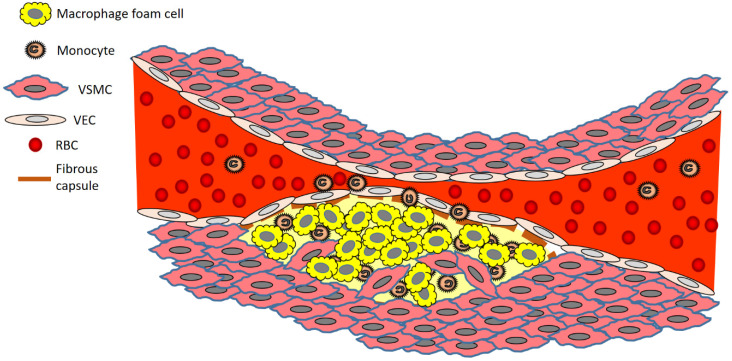
Brief overview of atherosclerosis development. Key: RBC, red blood cell; VEC, vascular endothelial cell; VSMC, vascular smooth muscle cell.

**Table 1 pharmaceutics-14-00935-t001:** Summary of studies using chitosan nanoparticles In vitro in atherosclerosis research.

Study Mode	Chemical Constituents/Type of Chitosan/Formulation Process	Size (nm)/ Shape	Agent Carried	Major Findings	Future Expectation	Reference
**In vitro** ***	AP peptide HGC Stirring	314 Spherical	NIR fluorophore Cy5.5	In TNF-α- activated BAECs, bound more avidly than to non-activated cells, with NPs seen within cells	In vivo evaluation was performed (Table 2)	[76]
**In vitro**	PAA Chitosan Electrostatic self-assembly	102 Spherical	EGCG	Improved stability in GI simulated fluids	In vivo evaluation was performed (Table 2)	[77]
**In vitro**	Cationic chitosan derivative Anti-VCAM-1 or anti-selectin Co-precipitation	91 Spherical	Fe^2+^ Fe^3+^	Specific interaction of SPIONs with aortic endothelial cells of db/db mice grown in culture	In vivo evaluation was performed (Table 2)	[78]
**In vitro**	TPP Pegylated chitosan Ionotropic gelation	<200 Spherical	RST	Sustained release of drug was observed in phosphate buffer pH 7.4	In vivo evaluation was performed (Table 2)	[79]
**In vitro**	PFH PLGA DS Chitosan Stirring	375 Spherical	Fe_3_O_4_	Avidly bound to and endocytosed by activated macrophages Induced apoptosis via LIFU in activated macrophages Aortic arch incubated with NPs post-LIFU developed holes, and the fibre cap was no longer intact	In vivo evaluation was performed (Table 2)	[80]
**In vitro**	TPP Pegylated chitosan Ionic gelation	150 Spherical	miR-33	Transferred miR-33 to macrophages and reduced the expression of ABCA1 Reduced cholesterol efflux to ApoA1	In vivo evaluation was performed (Table 2)	[81]
**In vitro**	Anti-CD47 antibody HA Chitosan Stirring	500 Roughly spherical	Cy5.5	Efficiently adsorbed to the surface of VECs	In vivo evaluation was performed (Table 2)	[82]
**In vitro**	Oxi-COS Extrusion	149 Spherical	Nile red	NPs responded to presence of ROS	In vivo evaluation was performed (Table 2)	[83]
**In vitro**	TPP Tween-80 Chitosan Ionotropic gelation	ND Spherical	RST	Initial burst release of 11.89%, then gradual sustained release of drug (~88.11%) over 48 h in PBS, pH 7.4	Evaluation in cells was not performed In vivo evaluation was performed (Table 2)	[84]
**In vitro**	TPP Poloxamer-188 Chitosan Ionic gelation	105 Roughly spherical	LST RST	Initial burst release of 15.24% (RST) and 13.98% (LST), then sustained release of 98.95% (RST) and 99.67% (LST) respectively over 14h in PBS, pH 7.4	Evaluation in cells was not performed In vivo evaluation was performed (Table 2)	[85]
**In vitro**	TPP Chitosan Electrostatic self-assembly	100 Heteroge- neous	Iron oxide Cerium oxide	Able to scavenge ROS Reduce the ROS level of LPS-stimulated macrophage Addition of TPP enhances cytotoxicity Able to act as contrast agent in macrophages as detected by MRI	In vivo evaluation not performed	[86]
**In vitro**	Lecithin Sodium monostearate Capric caprylic triglyceride scFv-antiLDL (-) Chitosan Coated surface	117 Spherical	Zinc	NPs internalised by phagocytosis and pinocytosis Macrophages exhibited lower LDL (-) uptake There was reduced mRNA and protein levels of IL1B and MCP1 induced by LDL (-)	In vivo evaluation was performed (Table 2)	[87]

**Key:***ABC1*, ATP binding cassette subfamily A member 1; *ApoA1*, apolipoprotein A1; *ApoE*, apolipoprotein E; *AT*, atorvastatin; *BAEC*, bovine aortic endothelial cells; *CD47*, cluster of differentiation 47; *CETP*, cholesteryl ester transfer protein; *CYP7A1*, cytochrome P450 7A1; *DS*, dextran sulphate; *EGCG*, epigallocatechin gallate; *FITC*, fluorescein isothiocyanate; *GPx*, glutathione peroxidase; *GSH*, glutathione; *HA*, hyaluronic acid; *IL1β*, interleukin-1β; *LDLr-/-*, low-density lipoprotein receptor-deficient; *LIFU*, low-intensity focused ultrasound; *LPS*, lipopolysaccharide; *LST*, lovastatin; *MCP-1*, monocyte chemoattractant protein-1; *MM*, macrophage membrane; *MRI*, magnetic resonance imaging; *ND*, not determined; *NIR*, near infrared; *NO*, nitric oxide; *NP*, nanoparticle; *Oxi-COS*, oxidation-sensitive chitosan oligosaccharide; *PAA*, polyaspartic acid; *PBS*, phosphate buffered saline; *PFH*, perfluorohexane; *PLGA*, poly (lactic-co-glycolic acid); *ROS*, reactive oxygen species; *RST*, rosuvastatin; *scFv-anti-LDL (-)*, single-chain fragment variable (scFv) reactive to LDL (-); *SOD*, superoxide dismutase; *SPIONs*, superparamagnetic iron oxide nanoparticles; *TNF-α*, tumour necrosis factor-α; *TPP*, tripolyphosphate; *VEC*, vascular endothelial cell. ** in case where no cell culture evaluation was performed, results from release studies are presented instead.*

**Table 2 pharmaceutics-14-00935-t002:** Summary of studies using chitosan nanoparticles In vivo in atherosclerosis research.

Study Mode	Chemical Constituents/Type of Chitosan/Formulation Process	Size (nm)/ Shape	Agent Carried	Delivery Route and Major Findings	Future Directions	Reference
**In vivo**	AP peptide HGC Stirring	314 Spherical	NIR fluorophore Cy5.5	Intravenous NPs bound better to atherosclerotic lesions in a low-density lipoprotein receptor-deficient (LDLr-/-) atherosclerotic mouse than to such lesions in a normal mouse	Examine whether these NPs are able to detect disease after a shorter period (less than 6 h post-administration) in mice Examine whether these NPs can ferry therapeutic agents to the lesion sites? Examine whether there are any side effects to continued usage of these NPs beyond the 6 h tested here	[76]
**In vivo**	pCETP Chitosan Complex coacervation	340 Spherical	pCETP	Cholesterol-fed rabbits intranasally immunised (6 times) Significant antibody generation against CETP detected Percentage aortic lesions were lower	Preclinical safety evaluation over longer time Phase I clinical trial to determine safety and serological titre of CETP antibodies If above is satisfactory, a phase II clinical trial to see if beneficial effects are obtained in atherosclerotic patients	[88]
**In vivo**	SP-2 HGC Self-assembly	315 Spherical	NIR fluorophore Cy5.5	NPs were delivered intraventricular to atherosclerotic plaques In vivo	The ability of these NPs to deliver anti-atherosclerotic agents Follow the delivery more than 1 hr post-NP-administration	[89]
**In vivo**	PAA Chitosan Electrostatic self-assembly	102 Spherical	EGCG	Oral administration reduced lipid burden similar to simvastatin	As data looked quite promising, a clinical trial could be performed	[77]
**In vivo**	Cationic chitosan derivative Anti-VCAM-1 or anti-selectin Co-precipitation	91 Spherical	Fe^2+^ Fe^3+^	Selective delivery to the aortic arch of ApoE/LDLR-/- mice	Evaluate safety of NPs over longer time Can the SPIONs be used to deliver therapeutic agents	[78]
**In vivo**	TPP Pegylated chitosan Ionotropic gelation	<200 Spherical	RST	Oral versus intravenous There was a greater lipid-lowering capability of RST when encapsulated	Perform a longer-term study and repeat dosing with NPs to assess efficacy and potential toxicity	[79]
**In vivo**	PFH PLGA DS Chitosan Stirring	375 Spherical	Fe_3_O_4_	Intravenous NPs could be targeted to the aortic plaque by inclusion of DS as determined via MRI Led to apoptosis of macrophages in plaques under LIFU NPs did not elicit toxic effects in heart, liver, spleen, lung, or kidney	Test the ability of NPs to completely eradicate plaques Observe if plaque removal causes any secondary emboli to form	[80]
**In vivo**	TPP Pegylated chitosan Ionic gelation	150 Spherical	miR-33	Subcutaneous Transferred miR-33 to macrophages and reduced the expression of ABCA1	Decreased RCT to the plasma, liver, and faeces ABCA1 expression and cholesterol efflux into the RCT pathway were improved	[81]
**In vivo**	Anti-CD47 antibody HA Chitosan Stirring	500 Roughly spherical	Cy5.5	Intravenous Distributed around the atherosclerotic plaque	Deliver therapeutic agents to the plaques	[82]
**In vivo**	Oxi-COS Extrusion	149 Spherical	AT	Intravenous AT-NPs/MAs showed a higher accumulation rate in the plaque tissue than that of MM-AT-NPs MM-AT-NPs exhibited hints of better therapeutic efficacy than AT-NPs/MAs	Examine whether cell membranes derived from other immune cells, including neutrophil, T cells, and B cells can also be utilised for drug delivery against atherosclerosis	[83]
**In vivo**	TPP Tween-80 Chitosan Ionotropic gelation	ND Spherical	RST	Route not mentioned Drug-loaded NPs significantly lowered blood lipid levels compared to pure drug Attenuated calcification of various valve tissues	Assess plaque area and stability	[84]
**In vivo**	TPP Poloxamer-188 Chitosan Ionic gelation	105 Roughly spherical	LST RST	Serum uric acid levels were lowered LST/RST-loaded NPs significantly lowered blood lipid levels compared to LST-or RST-loaded NPs	Check plaques histologically for area and stability	[85]
**In vivo**	Selenium Chitosan Chemical reduction	65.8 Spherical	AT	Intragastric administration Reduced atherosclerotic lesions in ApoE-/-mice Inhibited hyperlipidaemia by suppressing hepatic cholesterol and fatty acid metabolism Decreased oxidative stress by enhancing SOD, GPx and GSH activity Liver CYP7A1 mRNA levels were increased Increased serum NO levels Decreased serum TNF-α levels		[90]
**In vivo**	Lecithin Sodium monostearate Capric caprylic triglyceride scFv-anti-LDL (-) Chitosan Coated surface	117 Spherical	Zinc	Intravenous Reduced size of the atherosclerotic plaques	Delivery of other therapeutic, diagnostic or theranostic agents	[87]

**Key:***ABC1*, ATP binding cassette subfamily A member 1; *ApoA1*, apolipoprotein A1; *ApoE*, apolipoprotein E; *AT*, atorvastatin; *BAEC*, bovine aortic endothelial cells; *CD47*, cluster of differentiation 47; *CETP*, cholesteryl ester transfer protein; *CYP7A1*, cytochrome P450 7A1; *DS*, dextran sulphate; *EGCG*, epigallocatechin gallate; *FITC*, fluorescein isothiocyanate; *GPx*, glutathione peroxidase; *GSH*, glutathione; *HA*, hyaluronic acid; *IL1β*, interleukin-1β; *LDLr-/-*, low-density lipoprotein receptor-deficient; *LIFU*, low-intensity focused ultrasound; *LPS*, lipopolysaccharide; *LST*, lovastatin; *MCP-1*, monocyte chemoattractant protein-1; *MM*, macrophage membrane; *MRI*, magnetic resonance imaging; *ND*, not determined; *NIR*, near infrared; *NO*, nitric oxide; *NP*, nanoparticle; *Oxi-COS*, oxidation-sensitive chitosan oligosaccharide; *PAA*, polyaspartic acid; *PBS*, phosphate buffered saline; *PFH*, perfluorohexane; *PLGA*, poly(lactic-co-glycolic acid); *ROS*, reactive oxygen species; *RST*, rosuvastatin; *scFv-anti-LDL(-)*, single-chain fragment variable (scFv) reactive to LDL(-); *SOD*, superoxide dismutase; *SPIONs*, superparamagnetic iron oxide nanoparticles; *TNF-α*, tumour necrosis factor-α; *TPP*, tripolyphosphate; *VEC*, vascular endothelial cell.

## Data Availability

Not applicable.

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
