# Peer review of "Chitosan Nanoparticles in Atherosclerosis—Development to Preclinical Testing"

_pharmaceutics, 2022, doi:10.3390/pharmaceutics14050935_

Round 1

Reviewer 1 Report

Application of chitosan-based nanoparticle delivery systems in the treatment of atherosclerosis can be considered as an  area of research that is still in its infancy So it is important to select the most effective and promising directions of investigations where breakthrough results are able to accelerate the progress. So the review of this area of research is timely and potentially useful.

Some publications are described very briefly or simply only mentioned in the text without evaluation their potential impact.

The effect of chitosan on molecular mechanisms and  role of platelets in age-related diseases could be analyzed.

It would be beneficial to describe in more detail future directions.

It would beneficial to reformulate the statement in lines 37-39 taking into account a number of different possible understandings of atherosclerosis development in addition to the proposed in cited publication. 

Author Response

Reviewer #1

Application of chitosan-based nanoparticle delivery systems in the treatment of atherosclerosis can be considered as an  area of research that is still in its infancy So it is important to select the most effective and promising directions of investigations where breakthrough results are able to accelerate the progress. So the review of this area of research is timely and potentially useful.

Some publications are described very briefly or simply only mentioned in the text without evaluation their potential impact.

Please note that Tables 1 and 2 provide the major findings from all the studies evaluating the applicability of chitosan nanoparticles in vitro and in vivo. Providing them again in the text would take up valuable space in the journal, which was the reason why we presented it this way from the outset.

The effect of chitosan on molecular mechanisms and  role of platelets in age-related diseases could be analyzed.

We would like to point out that the manuscript was not intended to discuss molecular mechanisms of chitosan, but the use of chitosan in drug delivery nanoparticulate systems. To satisfy the reviewer’s wish, but to keep strictly within the scope of the theme of this paper, we have searched the literature thoroughly for papers describing the interaction of chitosan on platelets as it relates to atherosclerosis. Disappointingly, we could not find a single reference on the matter to include in our review article, despite not restricting our search to ‘chitosan nanoparticles’ but opening it up to any form of chitosan.

It would be beneficial to describe in more detail future directions.

We agree with the reviewer, and have now included a ‘Future perspectives’ section which discusses several future directions the research area can take. Please note that both Tables 1 and 2 also provide examples of ample avenues for future research in this area for the reader.

It would beneficial to reformulate the statement in lines 37-39 taking into account a number of different possible understandings of atherosclerosis development in addition to the proposed in cited publication. 

This is a good suggestion, and we have now put a few recent comprehensive citations for the reader to follow up on, as the area of atherosclerosis is a large and complex one, and cannot be justifiably covered in a small section. As this was also not the focus of this review, we did refrain from writing too much on it at the outset.

Reviewer 2 Report

This review article aims to describe the application of chitosan nanoparticles in the treatment of atherosclerosis. However, the organization needs to be more logic and clear. Significant revision is required before further consideration.

Specific comments:

  1. Nanoparticles is a wide concept, it can have different types and prepared with different technologies, it is not clearly described in this review article, and all the information are mixed together.
  2. Chitosan can be used as a homopolymer or used after modification, also this is not clearly described.
  3. The nanoparticles can be administrated via different route, this is closely related to the therapeutic effect, but no related information is clearly presented here.
  4. Figure 2 does not really make sense. Suggest delete.
  5. Add a “Discussion” section is not appropriate in a review article. Instead, expert opinion and future perspectives can be added.

Author Response

Reviewer #2

  1. Nanoparticles is a wide concept, it can have different types and prepared with different technologies, it is not clearly described in this review article, and all the information are mixed together.

To address this concern, we have included the type of formulation process and type of nanoparticle formulated and assessed in all the studies reviewed in Tables 1 and 2. 

  1. Chitosan can be used as a homopolymer or used after modification, also this is not clearly described.

This information is now provided for all studies reviewed in Tables 1 and 2.

  1. The nanoparticles can be administrated via different route, this is closely related to the therapeutic effect, but no related information is clearly presented here.

We thank the reviewer for pointing this out. Administration routes are now presented for all in vivo studies reviewed in Table 2.

  1. Figure 2 does not really make sense. Suggest delete.

As advised, Figure 2 has been removed.

  1. Add a “Discussion” section is not appropriate in a review article. Instead, expert opinion and future perspectives can be added.

As advised, a ‘Future perspective’ section is now included.

Reviewer 3 Report

The manuscript entitled “Chitosan Nanoparticles in Atherosclerosis – Development to Preclinical Testing” comprehensively discusses the use of chitosan nanoparticles for the treatment of atherosclerotic plaques. It is written well and fluently.

Just a few remarks:

In general, the text should be edited: there are parts highlighted in yellow (line 141-144) especially in the references.

The caption of figure 1 is to be summarized, I would insert the explanation in the text.

In Line 137 I suggest to include these 2 papers investigating chitosan nanopartcles.

Russo, N. Gaglianone, S. Baldassari, B. Parodi, I. Croce, A.M. Bassi, S. Vernazza, Caviglioli. Chitosan-clodronate nanoparticles loaded in poloxamer gel for intra-articular administration. Colloids and Surfaces B: Biointerfaces 143 (2016) 88–96.

Russo, N. Gaglianone, S. Baldassari, B. Parodi, S. Cafaggi, C. Zibana, Donalisio, V. Cagno, D. Lembo, G. Caviglioli Preparation, characterization and in vitro antiviral activity evaluation of foscarnet-chitosan nanoparticles. Colloids and Surfaces B: Biointerfaces 118 (2014) 117–125

Author Response

Reviewer #3

The manuscript entitled “Chitosan Nanoparticles in Atherosclerosis – Development to Preclinical Testing” comprehensively discusses the use of chitosan nanoparticles for the treatment of atherosclerotic plaques. It is written well and fluently.

Just a few remarks:

In general, the text should be edited: there are parts highlighted in yellow (line 141-144) especially in the references.

We apologise for this oversight, and have corrected this throughout the manuscript. Thanks to the reviewer for picking this up.

The caption of figure 1 is to be summarized, I would insert the explanation in the text.

We have now done this as advised.

In Line 137 I suggest to include these 2 papers investigating chitosan nanopartcles.

Russo, N. Gaglianone, S. Baldassari, B. Parodi, I. Croce, A.M. Bassi, S. Vernazza, Caviglioli. Chitosan-clodronate nanoparticles loaded in poloxamer gel for intra-articular administration. Colloids and Surfaces B: Biointerfaces 143 (2016) 88–96.

Russo, N. Gaglianone, S. Baldassari, B. Parodi, S. Cafaggi, C. Zibana, Donalisio, V. Cagno, D. Lembo, G. Caviglioli Preparation, characterization and in vitro antiviral activity evaluation of foscarnet-chitosan nanoparticles. Colloids and Surfaces B: Biointerfaces 118 (2014) 117–125

These two references have been added as suggested.

Round 2

Reviewer 1 Report

I agree with the corrections in the text

Reviewer 2 Report

It can be accepted now.